# Epigenetic Risks of Medically Assisted Reproduction

**DOI:** 10.3390/jcm11082151

**Published:** 2022-04-12

**Authors:** Romualdo Sciorio, Nady El Hajj

**Affiliations:** 1Edinburgh Assisted Conception Programme, Royal Infirmary of Edinburgh, Edinburgh EH16 4SA, UK; 2College of Health and Life Sciences, Hamad Bin Khalifa University, Doha P.O. Box 34110, Qatar; nelhajj@hbku.edu.qa

**Keywords:** human in vitro fertilization, assisted reproductive technology, epigenetics, imprinting disorders

## Abstract

Since the birth of Louise Joy Brown, the first baby conceived via in vitro fertilization, more than 9 million children have been born worldwide using assisted reproductive technologies (ART). In vivo fertilization takes place in the maternal oviduct, where the unique physiological conditions guarantee the healthy development of the embryo. During early embryogenesis, a major wave of epigenetic reprogramming takes place that is crucial for the correct development of the embryo. Epigenetic reprogramming is susceptible to environmental changes and non-physiological conditions such as those applied during in vitro culture, including shift in pH and temperature, oxygen tension, controlled ovarian stimulation, intracytoplasmic sperm injection, as well as preimplantation embryo manipulations for genetic testing. In the last decade, concerns were raised of a possible link between ART and increased incidence of imprinting disorders, as well as epigenetic alterations in the germ cells of infertile parents that are transmitted to the offspring following ART. The aim of this review was to present evidence from the literature regarding epigenetic errors linked to assisted reproduction treatments and their consequences on the conceived children. Furthermore, we provide an overview of disease risk associated with epigenetic or imprinting alterations in children born via ART.

## 1. Introduction

Over the past 40 years, the use of ART for infertility treatment has been continuously on the rise and has resulted in the birth of more than 9 million children globally [1,2]. The number of couples facing infertility problems has steadily increased over the last decades, particularly since a growing number of individuals are postponing the desire to have children further into older age. Many of those couples ultimately need in vitro fertilization (IVF) to be able to conceive a baby [3]. Nowadays, nearly 3.3 million ART cycles are performed annually, resulting in over 500,000 deliveries worldwide [1]. ART procedures are considered relatively safe; however, in the last decade, novel concerns have been raised due to increased prevalence of epigenetic errors and imprinting defects in ART-born children [4]. This was first observed in cattle and sheep, where incidence of large offspring syndrome (LOS) increased following transfer of in vitro fertilized embryos [5]. In 2001, Young et al. reported that epigenetic alterations in *IGF2R* was responsible for LOS following embryo culture in sheep [6]. Epigenetic alterations in various imprinted genes were also observed in preimplantation mouse embryos cultured in M16 or Whitten’s medium [7]. In vivo fertilization takes place in the oviduct, which is a natural environment with optimal physiological conditions including all the metabolic requirements for early embryo development. Even though embryology laboratories try to mimic those natural conditions to the best extent possible, during in vitro fertilization, the embryo is exposed to five or six days of diverse environmental conditions (Figure 1) [8]. Since about 3–5% of children are conceived following ART cycles [1], it is important to determine the potential negative effects of the procedure on the conceived baby. Epidemiological data revealed increased incidence of low and very low birth weight in ART-born babies following fresh embryo replacement [9]. Similar results were recently published by Sunkara et al., who analyzed UK registry data (Human Fertilization and Embryology Authority, HFEA) from 1991 to 2016 including about 117,000 singleton live births following ART. The authors showed that the causes of infertility had a negative impact on preterm birth and low birth weight following fresh embryo transfer [10]. However, the opposite scenario was reported following frozen-thawed embryo transfer (FET) in ART. A large study performed by Terho et al. suggested that FET is linked with higher birth weights and higher risk of large-for-gestational-age [11]. In 2002, a case report [12] was published describing two unrelated patients with Angelman syndrome with sporadic imprinting defects following intracytoplasmic sperm injection (ICSI). A year later, DeBaun et al. reported increased incidence of Beckwith–Wiedemann syndrome with imprinting alterations in *H19* and *LIT1* in children born after ART [13]. Subsequently, several studies tried to determine possible culprits behind the observed epigenetic errors including controlled ovarian stimulation (COS), in vitro oocyte maturation, intracytoplasmic sperm injection (ICSI), in vitro embryo culture, couple infertility, and more recently, preimplantation embryo manipulation for genetic assessment.

## 2. Epigenetics in Development and Imprinted Genes

In 1942, Conrad Waddington highlighted the importance of environmental interactions with genes during early stages of embryo development. Although at that time, only limited information was available about the mechanisms of early embryogenesis, Waddington emphasized the importance of studying features that control embryo development that can mediate the correlations between genotype and phenotype. Waddington introduced the term “*Epigenetics*”, which he described as the “the branch of biology that studies the causal interactions between genes and their products which bring the phenotype into being” [14]. Epigenetic regulation is essential for normal mammalian development and is described as the study of heritable changes in gene function that are not associated with changes to the DNA sequence itself [15]. In mammals, two waves of epigenetic reprogramming occur during development that reset epigenetic marks in germ cells and preimplantation embryos. During early embryogenesis, epigenetic marks are reprogrammed to prepare the embryo for development; however, parental-specific DNA methylation patterns at imprinted genes are maintained. The second phase occurs during germ cell development when primordial germ cells (PGCs) enter the fetal gonadal ridge. Here, DNA methylation patterns are globally erased including marks at imprinted genes. Parental imprinting marks are later established during germ cell differentiation with distinct imprints in male and female germ cells. During reprogramming, the epigenome is highly susceptible to external and internal cues that can alter the reprogramming process and induce long-term disease risk in the future generation [16,17]. One of the most studied epigenetic modifications is DNA methylation [18], where a methyl group is added at the 5′ carbon position of the cytosine pyrimidine ring in the context of CG dinucleotide (CpG sites) [19]. Those epigenetic modifications are maintained by daughter cells throughout cell divisions by DNA methyltransferase 1 (DNMT1) [20]. Epigenetic modifications are crucial in regulating gene expression during embryo development, whereby any disruption to epigenetic states during this sensitive time window can lead to future consequences for development and disease [21,22]. Genomic imprinting is an epigenetic process resulting in monoallelic expression of either the maternally or the paternally inherited allele. This mechanism of parent-of-origin-specific expression is restricted to a limited number of ~200 imprinted genes described in humans [23,24]. Genomic imprinting has been mainly reported in eutherian mammals; however, similar phenomena were identified in flowering plants and in some insects indicating independent evolutionary origins [25]. Imprinted genes are regulated by cis-acting elements known as imprinting control regions (ICRs). For example, in the *H19-Igf2* locus, the ICR located upstream of *H19* along with enhancers controls the expression of *H19* from the maternal allele and of the insulin-like growth factor (*IGF2*) gene from the paternal allele [26,27]. This exclusive monoallelic expression is controlled by specific epigenetic marks and regulatory elements such as DNA methylation, histone modifications, long non-coding RNA (lncRNA), and CCCTC binding factor (CTCF)-mediated boundaries [28]. The parental-specific imprints established in the germ line escape epigenetic reprogramming in preimplantation embryos, where imprinted genes play an important role in early development [29] and are essential for the regulation of energy balance between the mother and the developing fetus [30]. In humans, genetic mutations, copy number aberrations, and epigenetic alterations affecting imprinted genes have been linked to a number of disorders, e.g., Beckwith–Wiedemann syndrome (BWS), Angelman syndrome (AS), Silver–Russell syndrome (SRS), and Prader-Willi syndrome (PWS), Ref [31] characterized by clinical features affecting development, metabolism, and growth.

## 3. Epigenetic Alterations and Imprinting Disorders in ART

Following fertilization, the zygote develops into a structure called the “blastocyst” (Figure 2). At this stage, the embryo encloses about 150 or 200 cells differentiated into two types: the trophectoderm (TE), an epithelial sheet surrounding the fluid filled cavity (i.e., the blastocoele) and the inner cell mass (ICM), a group of cells attached to the inside of the trophectoderm that eventually give rise to the fetus. TE cells facilitate implantation into the uterine lining and form extraembryonic tissues including the placenta. During early development, embryonic cells are guided toward their future lineages through epigenetic reprogramming and subsequent re-establishment of cell-type-specific epigenetic signatures. This corresponds to the period when gametes and embryos are being in vitro manipulated and cultured inside the embryology laboratory. Therefore, such artificial intrusions during this critical time window might lead to epigenetic aberrations in the resultant offspring (Figure 1 and Figure 3). Several studies reported imprinted loci to be vulnerable to external environmental cues during in vitro embryo culture. For example, *KvDMR1* has been observed to be abnormally methylated in ART-related BWS in humans [32,33] and hypomethylated in ART-produced bovine conceptuses with LOS [34]. Several studies have also shown that ART-related procedures including COS, ICSI, and embryo manipulation might induce epigenetic abnormalities [29,31,35]. A systematic review published by Lazaraviciute et al. compared the incidence of imprinting disorders and DNA methylation alterations at key imprinted genes in children conceived via ART versus those conceived naturally. A total of 18 papers were included in this review, and the combined odds ratio (95% confidence intervals) for the incidence of imprinting disorders in children conceived through ART was 3.67 in comparison to spontaneously conceived children. The authors concluded that an increased risk of imprinting disorders occurs in babies born via IVF and ICSI; nevertheless, there was limited evidence for a link between epigenetic alterations at imprinted genes and ART [36]. Another review summarizing data from eight studies on BWS and ART reported a significant positive association between IVF and ICSI procedures and BWS with increased relative risk of about 5.2 times (95% CI 1.6–7.4) [37]. However, the authors did not observe an association for either AS or PWS with IVF and ICSI, but rather a positive association with fertility problems. Regarding SRS, the number of children born following ART was small (*n* = 13); therefore, probable significance for SRS incidences could not be inferred. A more recent epidemiological study investigated the risk of imprinting disorders in IVF children born in Denmark and Finland, where the authors compared the incidence rate of PWS, SRS, BWS, and AS in ART-conceived babies in Denmark (*n* = 45,393 born 1994–2014) and Finland (*n* = 29,244 born 1990–2014). They observed an increased odds rate for BWS (OR 3.07, 95% CI: 1.49–6.31) in ART-conceived children; however, no significant difference was evident for PWS, SRS, and AS [38]. Similarly, a nation-wide study in Japan found a 4.46-fold increase in BWS and an 8.91-fold increase in SRS following ART including several with aberrant DNA methylation at imprinted genes [39]. The effect of altered epigenetics marks and epimutations on human health is just beginning to be understood. Further research in this area is needed help clarify whether ART-induced epigenetic changes affect growth, development, and health of future offspring. In the next sections, we discuss specific procedures applied during ART treatments to provide examples on how certain treatments may lead to epigenetic alterations.

## 4. Controlled Ovarian Stimulation in ART

Ovarian stimulation is one procedure likely responsible for epigenetic aberrations in the oocyte and embryo [40]. COS may lead to the selection of poor quality oocytes that are usually excluded in a natural cycle, and those oocytes might induce perturbed genomic imprinting during the early stage of embryo development and later in the placenta [41,42]. Medical records of women who gave birth to children with BWS following ART revealed ovarian stimulation medication as the only common factor among those patients [43]. Each month, the human ovaries typically produce a single dominant follicle which ovulates and releases a single oocyte. To increase the number of fertilized oocytes and improve IVF outcome, COS is applied using exogenous gonadotropins to stimulate the ovary and promote multifollicular development yielding multiple oocytes. Typically, a pharmacological dose of FSH is used to induce the growth of multiple follicles. As follicles grow and reach a specific width, LH is administered to produce the mid-cycle LH surge, which promotes oocyte maturation and later ovulation. Oocyte retrieval is precisely timed following LH administration to retrieve mature oocytes prior to ovulation. LH exposure initiates meiosis and leads to oocyte maturation from the immature “metaphase I” (MI) stage to the mature “metaphase II” (MII) stage of development. During this time, the first polar body is extruded and the oocyte reaches the metaphase II stage, which indicates its competence to be fertilized [44]. Following ovulation, the rest of the follicle forms the corpus luteum, which produces high levels of progesterone to prepare the endometrium for the process of embryo implantation. Since the expected number of oocytes is low in patients with reduced ovarian reserve, several strategies mainly based on increased gonadotropin dose have been applied to collect more oocytes. In certain cases, it is only possible to retrieve immature oocytes after COS where in vitro maturation might be adopted to obtain matured MII oocytes. Culture systems for in vitro maturation of human oocytes holds great potential but is still considered experimental for clinical use in ART [45]. In the last decade, there has been a growing concern over an association between COS and epigenetic aberrations in oocytes and embryos, which further increases the risk of imprinting disorders in the offspring [46]. Indeed, DNA methylation analysis of imprinted genes revealed aberrations in *PEG1, KCNQ1OT1,* and *ZAC* in oocytes collected following COS when compared to oocytes obtained after natural ovulation [47,48]. Furthermore, reports described DNA methylation alterations and expression changes in the *H19* imprinted control region in embryos obtained from superovulated oocytes [49]. Mature oocytes obtained following superovulation were shown to have conserved DNA methylation patterns at ICRs; however, methylation aberrations were detected in genes involved in glucose metabolism, nervous system development, mRNA processing, cell cycle, and cell proliferation [50]. This is in contrary to a genome-wide DNA methylation study in superovulated mouse oocytes, which showed minor methylation differences between superovulated versus naturally ovulated oocytes [51]. DNA methylation was also studied in embryos generated from superovulated oocytes, where superovulation was shown to interfere with the genome-wide DNA methylation reprogramming process that occurs during early embryogenesis [52]. Multiple superovulation cycles were also shown to have adverse effects on the structure and function of the ovaries, causing lower fertilization rate and decreased rate of early embryo development. In addition, repeated superovulation affected expression of pluripotency genes and led to aberrant histone modifications in early embryos and in the future offspring [53,54]. However, the effect of the ovarian superovulation on various epigenetic mechanisms are still to be fully elucidated. In animal models, reports have largely described that COS might alter the correct activities of DNA methyltransferases [53,54]. One of the first studies to determine that superovulation modifies expression levels of the DNMT proteins was published by Uysal et al. [55]. In this study, the authors compared DNMT protein levels in three groups (control, high dose, and normal dose of gonadotropins) and found that DNMT1, DNMT3A, and DNMT3B protein expression in the oocytes and developed embryos differed significantly when compared with controls. Similar data have been published by other groups confirming those results [53,54,56,57].

## 5. Fertilization Procedures: In Vitro Fertilization (IVF) and Intracytoplasmic Sperm Injection (ICSI)

There are two techniques used for oocyte fertilization in vitro: (1) the standard insemination where sperm and oocyte are placed together overnight in a culture dish for the sperm to fertilize the oocyte and (2) the intra-cytoplasmic sperm injection (ICSI) where an embryologist adopting an inverted microscope and a micromanipulator with a slim injection pipette collects and immobilizes a single sperm before slowly releasing it into the oocyte’s cytoplasm. ICSI was first performed by Palermo et al. in 1992 [58] and it was introduced in clinical practice without prior experimental testing or clinical validation in animal models. Since then, it has been one of the major advances in ART for infertile couples diagnosed with severe male factor infertility. Natural fertilization usually follows specific physiological events including natural sperm selection and capacitation as well as acrosome reaction and membrane fusion before the sperm nucleus is released into the oocyte cytoplasm. Nevertheless, all these processes that occur during fertilization are basically omitted when ICSI is applied [59]. The usage of ICSI is increasing recently, where the technique is even applied in couples with men having semen analysis within reference ranges. Currently, ICSI is the main insemination technique in several infertility centers and in the Middle East, it is adopted in ~96% of all ART cycles [60]. Several researchers have put forward the idea that imprinting errors may originate due to abnormal spermatogenesis, which are later transmitted to the embryo following ICSI. For example, DNA hypomethylation at the *H19* gene locus in sperm has been associated with oligozoospermia and azoospermia [61]. Similarly, Kobayashi et al. studied imprinting in sperm of 97 infertile men where they identified errors at paternally imprinted genes in 14.4% of patients and errors at maternally imprinted genes in 20.6% of patients. The majority of imprinting defects were in oligospermic men, which led the authors to conclude that infertile men with abnormal sperm parameters have an increased risk of transmitting incorrect imprints to their offspring [62]. Similarly, Marques et al. observed increased risk for *H19* hypomethylation in testicular spermatozoa from men with abnormal spermatogenesis, indicating a possible link between disruptive spermatogenesis and imprinting errors [63]. During ICSI, natural sperm selection is omitted where sperm from men with severe male factor infertility might lead to the transmission of imprinting errors to the offspring. Furthermore, sperm following testicular sperm extraction (TESE) from men with non-obstructive azoospermia have been also used in ICSI procedures. In contrast to the previously mentioned studies, several reports showed no increased risk of epigenetic alterations in children born following ART. For example, a retrospective cohort study measuring DNA methylation in the *PEG3*, *IGF2*, *SNRPN*, and *INS* genes as well as the long interspersed nuclear element I (LINE-1) observed no significant DNA methylation differences in ART-conceived children [64]. Another study by Rancourt et al. investigated methylation levels of *GRB10, MEST, H19, SNRPN*, *KCNQ1,* and *IGF2DMR0* where they found no association between epigenetic aberrations and ART [65]. Additional studies have similarly reported no significant global or imprint-specific differences when comparing children born following IVF, ICSI, and natural conception [66,67,68]. More recently, a genome-wide DNA methylation analysis could only identify DNA methylation changes of small effect size in cord blood of ICSI-born children including eight sites at imprinted control regions [69]. Following a targeted and genome-wide DNA methylation analysis, Barberet et al. found lower methylation levels in buccal smear DNA at the *H19/IGF2* DMR in ART children as well as higher *PEG3* DMR methylation. However, the authors could only observe lower DNA methylation levels at the *LINE-1* transposable elements when comparing ICSI children to their IVF counterparts [8]. Another study by Choux et al. investigated the relation between ART and DNA methylation alterations in imprinted genes. The authors analyzed DNA methylation and expression levels of three imprinted loci (*H19/IGF2, KCNQ1OT1,* and *SNURF DMRs*) in cord blood and placenta obtained at birth from 15 standard IVF and 36 ICSI singleton pregnancies versus their 48 spontaneously conceived counterparts. Results showed that DNA methylation levels of *H19/IGF2, KCNQ1OT1, LINE-1Hs,* and *ERVFRD-1* were significantly lower in IVF and ICSI placentas than in control placentas, while there was no difference for cord blood [70]. Recent studies have shown that the placenta is more susceptible to epigenetic alterations when compared to the embryo and can therefore be used as a proxy to measure early epigenetic alterations affecting the embryo [71,72,73,74]. For example, placentas from ICSI- but not IVF-born children were reported to have global *H3K4me3* differences when compared to natural conceptuses [75]. A comprehensive study by Choufani et al. examined placentas from singleton pregnancies in an ART group and matched controls enrolled in the Quebec-based Canadian 3D longitudinal cohort, where they observed outliers in placentas of ART conceptuses to be enriched for DNA hypomethylation at imprinted genes. Furthermore, they observed that paternal age and infertility further perturbed the placental epigenome of ART-born children [67]. They found hypomethylation at imprinted genes to be associated with lower *H3K9me3* (repressive) and higher *H3K4me2* (permissive) marks [76]. This is in line with other reports that identified age-related changes in the sperm epigenome that might be later transmitted to the offspring [77,78]. In addition, male obesity and paternal diet was associated with malleable changes in the sperm epigenome [79,80,81,82]. Recent evidence has shown that disruption to the paternal epigenome can induce male infertility and subsequently transfer epigenetic aberrations to the embryo and potentially to the offspring, especially when fertilization is achieved using ART or ICSI. An analysis published by Schon et al. observed an overall reduction in *H4* acetylation as well as alterations in *H4K20* and *H3K9* methylation in asthenoteratozoospermic men compared to normozoospermic samples [82]. Furthermore, a study by Vieweg et al. found that abnormal histone acetylation in gene promoters of infertile men is associated with insufficient sperm chromatin compaction, and this alteration could potentially be transmitted to the future offspring [83]. Similarly, other studies reported alterations in methylation imprints in sperm of men with abnormal sperm parameters as well as methylation differences at *ALu* repeats that could be even associated with ART outcome [84]. ICSI might increase the incidence of imprinting disorders, adversely affect embryo development, and eventually lead to adverse health consequences in the resulting children [83,85]. Several reports have challenged the extensive usage of ICSI as well as its advantages compared to traditional IVF [86]. As a result, the Practice Committee of the American Society for Reproductive Medicine (ASRM) has recently produced a committee opinion paper recommending against the extensive use of ICSI in couples undergoing MAR cycles without male factor infertility [87].

## 6. Epigenetic Alterations Following In Vitro Culture

Despite in vitro fertilization being routinely practiced in couples with infertility issues, the cause for the increased risk for perinatal problems in ART-conceived children is still poorly understood. Animal models have provided evidence suggesting that imprinting establishment in oocytes and embryos is sensitive to environmental changes. Several studies have described the effects of in vitro culture on gene expression in preimplantation embryos in different mammals [72,73,74,88,89,90]. Epigenetic marks necessary for optimal embryo development are acquired during gametogenesis (imprinting) and preimplantation embryo development. Correct establishment of epigenetic patterns is crucial for development; however, morphological assessment of gametes and/or embryo quality cannot identify epigenetic errors during ART treatment [91]. Several trials have shown disrupted methylation at a number of imprinted genes due to in vitro culture in certain media [7,49,92,93,94,95]. A comprehensive study by Schwarzer et al. analyzed IVF procedures and in vitro culture media versus in vivo controls. In total 5735 fertilized mouse oocytes were cultured in vitro or in the female oviduct and scored for developmental parameters at the blastocyst stage (around 96 h). The authors reported that culture media might induce a wide range of changes in cellular, developmental, and metabolic pathways [96]. Similar results were observed by Gad et al. while investigating the effect of different culture media on the transcriptome profile of bovine preimplantation embryo development [97]. In humans, a handful of studies have explored the effects of culture media in preimplantation embryos. Kleijkers et al. cultured human embryos in two different culture media, where they observed differential expression of 951 genes involved in apoptosis, metabolism, protein processing, and cell cycle regulation diverged significantly when comparing blastocysts cultured in either G5 or human tubal fluid (HTF) [98]. Similarly, a more recent study reported differential expression of several genes between human cryopreserved embryos cultured using the same two media; however, expression differences were higher due to maternal age and developmental stage. The authors were not able to confirm whether the observed differences might be caused by confounding factors and concluded further research is needed to validate those results [99]. A randomized controlled trial compared DNA methylation at imprinted genes in IVF placentas from embryos cultured in HTF versus G5 medium, where no significant differences in DNA methylation were detected. Furthermore, no DNA methylation differences were observed when comparing IVF versus naturally conceived placentas, despite IVF placentas exhibiting a higher number of outliers [100]. A striking example of the negative effects of in vitro culture on embryo development was observed in cattle with LOS [5]. A study published by Chen et al. highlighted the concern that in vitro culture and ART induces misregulation of several imprinted genes in the kidney, brain, and liver of LOS fetuses. The magnitude of overgrowth in LOS fetuses is associated with the number of epigenetically altered imprinted genes [40].

## 7. Oxygen Tension

In vitro culture is thought to be one of the most important factors affecting epigenetic reprogramming as well as the developmental potential of embryos produced by ART. Since the 1950s, research has been conducted to determine the concentration of oxygen in the female reproductive tract. Historically, embryo culture has been performed at atmospheric oxygen levels of around 20%. Later, it was established that oxygen concentration in the female reproductive tract of mammalian species is between 2–8% [101], which indicates that embryos develop in vivo under low oxygen concentrations [101,102]. Several studies on mammals, including humans, suggested adverse effects of atmospheric oxygen levels on embryo development [103,104] as well as changes in the proteome [105], the transcriptome [106], and the epigenome of the embryo [29]. In the cytoplasm, oxidative stress resulting from the accumulation of reactive oxygen species (ROS) is likely a mechanism via which high oxygen concentration weakens the embryo, reducing its implantation potential and its capacity to generate a viable pregnancy. It has been proposed that in vitro culture of human embryos at reduced oxygen tension is an important feature to retain physiological evolution and increase reproductive competence. Indeed, there is plenty of evidence advocating in vitro culture of human embryos at 5% levels, rather than ambient oxygen, to improve pregnancy outcomes [105,106,107]. A recent prospective randomized multicenter study performed on 1563 oocytes confirmed that inclusion of antioxidants to the culture media significantly increases embryo viability, implantation, and pregnancy rates, possibly via oxidative stress reduction [108]. Similarly, the Cochrane Database review confirmed the results of several trials showing that in vitro culture of human embryos under conditions of low oxygen concentration improves ART outcomes [109].

## 8. In Vitro Culture and Human Birthweight

Birthweight is a useful and essential metric related to fetal growth and is suggested by some as a possible prognostic factor of long-term risk of metabolic disease. Low birthweight is known to be associated with increased rates of coronary heart disease as well as related disorders such as stroke, hypertension, and non-insulin dependent diabetes [110]. A study by Dumoulin et al. compared pregnancy rates and perinatal outcomes from singleton pregnancies born following 826 first IVF cycles, in which embryos were randomly cultured in two different sequential media. In total, 110 live-born singletons were analyzed where a significant difference in birthweight (3453 +/− 53 versus 3208 +/− 61 g, *p* = 0.003) adjusted for gestational age and sex was observed. This led the authors to conclude that in vitro culture of human embryos can affect the birth weight of live-born singletons [111]. This finding was confirmed by the same group in a separate report, where they studied a larger cohort of 294 live-born singletons [112]. Similarly, other groups reported comparable results to the previously mentioned studies [98,113,114,115]. IVF culture medium were also shown to be associated with postnatal weight changes during the first two years of life, suggesting that the early stage of human embryo development is susceptible to the external environment and that the culture medium might have long-term consequences [98,116]. On the other hand, a retrospective study published by Lin et al. comparing the effect of three commercially available culture media on the birthweight and length of newborns revealed no significant differences in mean birthweight [117]. Further studies using a different range of culture media also reported no significant differences in birthweight [118,119]. Despite conflicting results, the debate is still ongoing and no definite conclusion can be drawn. Therefore, it is essential to longitudinally follow-up IVF-born children and monitor their long-term growth, development, and health. Several other factors during in vitro culture might have an effect on birthweight such as the age of the culture media, storage time in the fridge or in the incubator [120], as well as the protein source and the used concentration [121]. Furthermore, one of the most debatable questions is related to culture period length, as well as to whether the embryo is transferred to the uterine cavity at the cleavage stage (day 2–3) or the blastocyst stage (day 5). This aspect has been investigated by Zhu et al. in a retrospective analysis of 2929 singletons, where the authors found that birthweight of singletons after blastocyst transfer was significantly higher than singletons from embryos at day three transfer (3465.31 ± 51.36 versus 3319.82 ± 10.04 g, respectively, *p* = 0.009) [122]. These questions were also addressed in a systematic review [123] that looked at several published human studies investigating the association between culture media and birthweight. The authors concluded that out of the 11 published studies, only six reported differences in birthweight while five observed no changes. As discussed earlier, epidemiological studies reported an increased incidence of low and very low birth weight in ART-born babies following fresh embryo transfer [9,10,11]. On the other hand, a different picture emerges following FET in ART. A recently published large-scale study have analyzed live-born singletons born in Denmark, Norway, and Sweden between the years 2000 and 2015. The authors correlated singletons born after FET (*n* = 17,500) to singletons born after fresh embryo transfer (*n* = 69,510) and natural conception (*n* = 3311.588). Results showed that birth weights were significantly higher after FET compared to fresh ET for both boys and girls [11]. Comparable results have been also published by Litzky et al. using data from registries in the United States and analyzing the impact of FET (*n* = 55,898) versus fresh embryo transfer (*n* = 180,184) on birth weight of singletons conceived via ART between 2007–2014. Results found that FET was correlated with, on average, a 142 g increase in birthweight compared with infants born after fresh embryo transfer (*p* < 0.001) [124].

## 9. Cardiometabolic Complications in ART-Conceived Children

In addition to fetal growth restriction, prematurity, and low birth weight, certain studies have reported a possible association between ART cycles and a slightly increased risk of cardiovascular diseases [125]. A study conducted in Sweden compared the presence of congenital malformations in 15,570 infants born following ART versus all infants born in Sweden between 2001–2007. This analysis revealed a slightly increased risk of congenital malformations, cardiovascular disease, neural tube defects, and esophageal atresia after IVF [126]. Similarly, a trial was performed in Australia to determine disease risk in children (at least 1 year of age) born following IVF treatment. Results from this study suggested an increase in the incidence of raised blood pressure, elevated fasting glucose, and higher total body fat composition in IVF offspring. Nevertheless, it is still debatable whether these potential associations are related to the ART procedure itself or prior genetic susceptibility in the children [127]. A separate study assessed systemic and pulmonary vascular function in 65 healthy children born after ART, where they reported a 30% higher (*p* < 0.001) systolic pulmonary artery pressure in ART versus naturally conceived children (*n* = 57) [128]. Similarly, von Arx et al. compared the cardiac function and pulmonary artery pressure in 54 healthy children conceived via ART versus 54 age- and sex-matched control children. In this study, they observed increased right ventricular dysfunction in children and adolescents conceived by ART under stressful conditions of high-altitude pressure and hypoxia [129]. This concern has been also investigated in twin pregnancies following ART. Multiple pregnancies are common following ART and are normally linked with increased adverse perinatal outcomes such as hypertensive disorders, gestational diabetes, and preterm birth. In a recent study, Valenzuela-Alcaraz et al. investigated the presence of fetal cardiac remodeling and disruption in ART twin pregnancies [130]. The authors found that in comparison to non-ART conceptuses, twin pregnancies following ART showed significant cardiac changes, predominantly affecting the right heart, such as dilated atria, more globular ventricles, and thicker myocardial walls, as well as reduced longitudinal motion (*p* < 0.001). This study confirmed aberrations that are similar to those observed in ART singletons. Additional studies have reported similar results, thus reinforcing the evidence of an increased risk for metabolic and cardiovascular diseases following ART [131,132,133]. However, Bi et al. recently reported that changes associated with cardiac morphology and function seems to be limited to ART fetuses and do not persist towards early infanthood [134]. A more recent study on the Growing Up in Singapore Towards healthy Outcomes (GUSTO) prospective cohort observed no changes in metabolic biomarkers in ART conceived singletons; however, those children were shorter, weighed less, and had lower blood pressure and reduced skinfold thickness compared to their naturally conceived counterparts at ~6–6.5 years of age [135]. A recently published prospective study compared socioeconomic, psychosocial, and clinical measures in ART-conceived singletons who were 22–35 years of age during the time of the study. It was reassuring that the authors did not observe increased risk of cardiometabolic, growth, or respiratory problems in young adults conceived via ART when compared to the non-ART group [136]. An elegant follow-up study by Novakovic et al. performed genome-wide DNA methylation analysis in Guthrie spots and whole blood DNA in the same cohort, where ART procedures were shown to be associated with DNA methylation alterations at birth that did not persist into adulthood [137].

## 10. Epigenetic Alterations and Preimplantation Genetic Testing Following ART

Embryo biopsy for preimplantation genetic testing for aneuploidies (PGT-A) or further aspects related to preimplantation diagnostics might also have effects on the epigenome of the offspring [138,139]. PGT-A is used to avoid the transfer of chromosomally abnormal embryos to reduce implantation failures and miscarriages and is normally advised for advanced maternal age (AMA), repeated implantation failure (RIF), and recurrent pregnancy loss (RPL) [140]. The genetic assessment is linked to the biopsy procedure, which in the early days was based on blastomere aspiration collected from a cleavage stage embryo (day three). The method uses the acid Tyrode solution to make a hole in the zona pellucida (ZP) for subsequent aspiration of the cell. Later, laser-assisted zona drilling and calcium magnesium free media was introduced, which allows for easier blastomere removal. Trophectoderm biopsy (TEB) was suggested in 1990 [141], which allows the collection of more genetic material (~5–10 cells) for improved diagnostic accuracy [140,141]. However, considerations on the safety of PGT-A have been until now not well considered, particularly issues related to sampling strategy (i.e., blastomere biopsy at the cleavage stage or trophectoderm biopsy), as well as the manipulation and change in culture media during the procedure [101,102,103,104]. Specific settings such as temperature, culture media pH, and a reduced physiologic 5% oxygen tension have an effect on embryo quality and can mediate epigenetic dysregulation [103,105,106]. Similarly, the approach used to dissect the zona pellucida might harm the embryo and impair its development [29,142]. Recently, PGT-A is moving to TEB, normally performed on days five and six or even on day seven in certain cases. Although there is still limited evidence favoring blastocyst transfer in ART [143], extended in vitro culture beyond the embryonic genome activation (EGA) stage might have negative effects on the embryo. Several review studies raised the alarm over an increased incidence of negative obstetric and perinatal outcomes from extended embryo culture, pointing to possible epigenetic alterations in the embryo [8,69,88,89,91,94]. However, due to the limited number of studies on human embryos, it is difficult to delineate whether epigenetic alterations arise due to infertility (Figure 4), follicular stimulation, or embryo culture per se [96,97,98,99]. An elegant study in the bovine model allowed embryos to develop in vivo up to the 2, 8, and 16 cell stage followed by in vitro culture until the blastocyst stage to separate the epigenetic alterations sourced in the different phases of embryo development. This study demonstrated that every step of in vitro culture before and during embryonic genome activation (EGA) was contributing to epigenetic alterations. However, the majority of changes were occurring around the EGA phase with far less alterations after genome activation [144]. Furthermore, the sensitivity of embryo culture to oxygen drives the attention to preimplantation embryo metabolism and to a possible role of the culture media in improving outcomes following PGT-A. Cytogenetic composition and health of human embryos in vitro have been shown to be associated with metabolism, where aneuploid human embryos were reported to carry significant changes in amino acid turnover as measured in their spent culture medium [145,146]. This opens the possibility to measure embryo metabolism as a biomarker to assess embryo quality and for monitoring the effect of culture conditions to reduce mitotic error rate. Nevertheless, as long as our understanding of human preimplantation embryo metabolism is limited, one has to carefully consider that any additional day of in vitro culture has the potential to negatively affect the embryo and induce epigenetic alterations.

## 11. Conclusions and Future Perspectives

ART procedures have helped millions of infertile couples in having children; however, several concerns remain regarding the safety of these techniques on the health and well-being of the offspring at birth and in later adult life. The main aim of this review was to provide an overview of epigenetic alterations associated with in vitro fertilization and culture of human embryos. Several studies in animal models as well as retrospective follow-up studies of ART-born babies have reported an increased risk of epigenetic errors particularly affecting imprinted loci. Nevertheless, there is still no conclusive evidence of a strong link between ART and epigenetic modifications as well as increased disease risk in later adult life. It is important to mention that manipulation of oocytes and embryos should be restricted to a minimum, or in other words, the advantage of a specific technique such as extended culture to the blastocyst stage or preimplantation genetic assessment must outweigh the potential negative effects. Unfortunately, many decisions in human-assisted reproduction are not based on conclusive evidence, since longitudinal studies with a follow-up over several decades are still very limited. Therefore, large-scale epidemiological studies to evaluate the implications of various ART techniques on the health and well-being of the offspring not only at the time of delivery but also during later adult life are urgently needed.

## Figures and Tables

**Figure 1 jcm-11-02151-f001:**
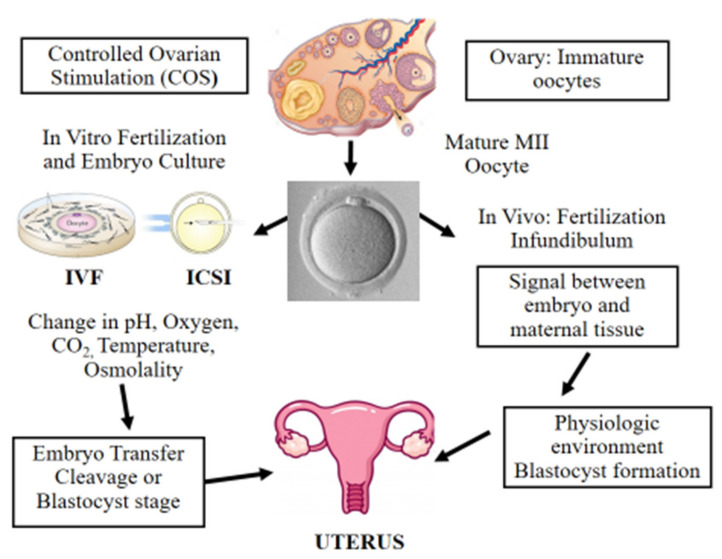
Scheme illustrating in vitro and in vivo fertilization. Controlled ovarian stimulation (COS) is used to promote follicle growth, maturation, and ovulation. ART adopts either IVF or ICSI for fertilization. Following fertilization, the preimplantation embryo is cultured in incubators, where suboptimal culture conditions such as pH, oxygen, temperature, and osmolality may affect its further development. Finally, the in vitro-produced embryo is transferred to the uterus at the cleavage or blastocyst stage. On the other hand, in vivo the female and male gametes interact together and the sperm fertilizes the oocyte in the infundibulum. Next, the developing embryo moves towards the uterus interacting with the female reproductive system in a physiologic and optimal environment.

**Figure 2 jcm-11-02151-f002:**
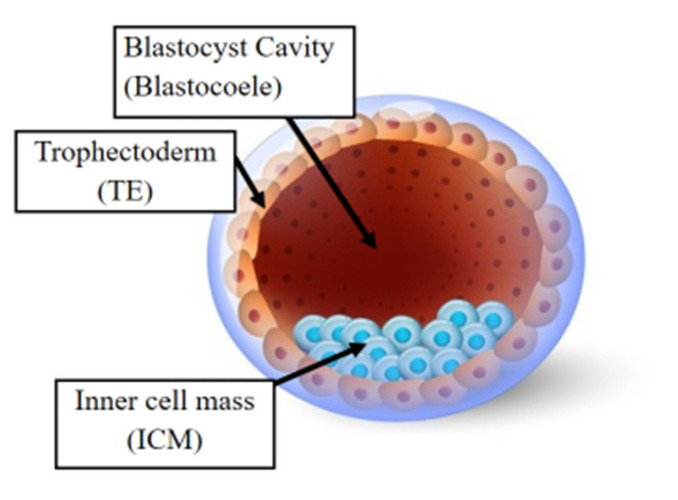
The human blastocyst. The structure comprises two differentiated cell types and a central cavity filled with fluid (blastocoel cavity). The inner cell mass (ICM) becomes the fetus and the trophectoderm (TE) cells later develop into the placenta.

**Figure 3 jcm-11-02151-f003:**
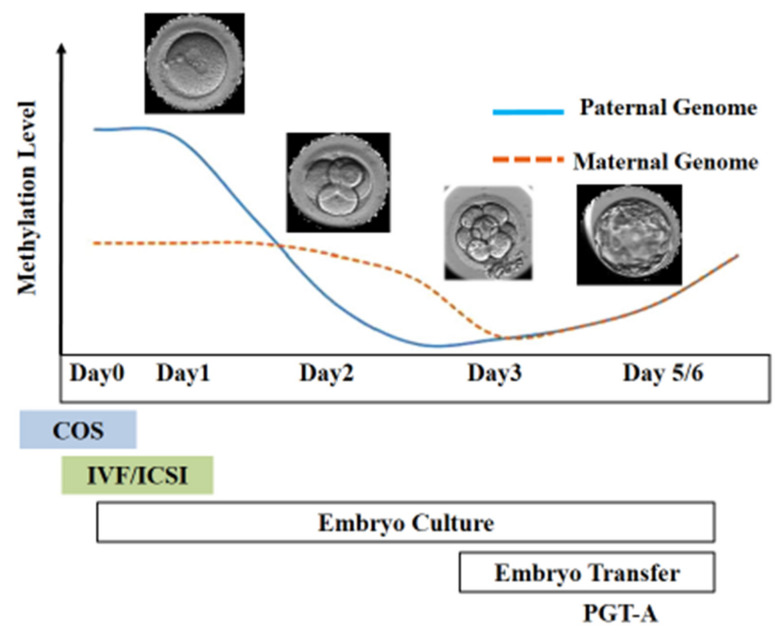
Epigenetic reprogramming during the early stage of embryo development. Post-fertilization, the paternal genome undergoes active demethylation, whereas the maternal genome is passively demethylated. The scheme illustrates the stage of development at which different ART techniques are employed.

**Figure 4 jcm-11-02151-f004:**
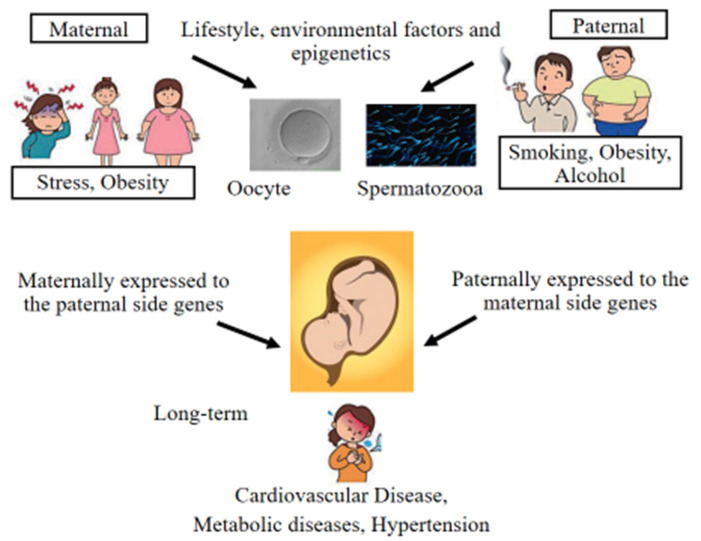
Paternal and maternal lifestyle prior to conception may affect sperm and oocyte epigenetic changes, offspring epigenetics, and phenotypic abnormalities, including increased risk of cardiovascular and metabolic disease in later life.

## Data Availability

No data are available.

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
