# Peer review of "Epigenetic Risks of Medically Assisted Reproduction"

_jcm, 2022, doi:10.3390/jcm11082151_

Round 1

Reviewer 1 Report

The article is well written and structured. It was performed an exhaustive  and interesting analysis of literature. 

Conclusion and Future Perspective (line 448): Paragraph number 11

I want to suggest improving the image quality.

Author Response

Thanks very much for the time you spent to review our paper, and thank you for the comments..

Please found here attached answers at your points.

Best Regards

Romualdo Sciorio

Reviewer 2 Report

"Epigenetic Risks of Medically Assisted Reproduction" is a review article that summarizes the literature regarding data related to epigenetic alterations in embryos, placentas, and offspring following ART procedures.

Overall, the manuscript is well-written and is well-organized in presenting the data, going step-by-step through various ART procedures and summarizing the available literature.

I do think it is a reasonable contribution to the literature and can be a helpful review, especially to someone new to the field.

However, I do think there are some areas in which the review can be improved:

  1. Overall, the authors highlight that there may be an effect of infertility itself on epigenetic changes and this is difficult to establish in human research (more easily established in animal models).  This distinction is well-made in the section regarding ICSI but is not emphasized sufficiently in other areas of the paper.  The authors may want to consider adding a section specifically discussing the epidemiologic data regarding infertility and ART as co-contributors to epigenetic changes, or they should make this distinction clearer in the latter sections of the paper which discuss neonatal outcomes (birthweight, cardiac outcomes).
  2. The authors cite that birthweight is lower in babies following ART (pate 2 line 50); however, they do not clarify that this distinction is specific to the type of ART cycle performed (fresh compared to frozen embryo transfer), nor do the authors discuss newer data discussing alterations in birthweight specific to programmed FET cycles, which would be relevant to this discussion.
  3. In section 4 (Superovulation and Epigenetics)- Page 5-6: Would suggest a discussion of the mechanism responsible for the effect of the superovulation on methylation, as it likely not due purely to oocyte selection alone but perhaps due to aberrations in follicular development due to the administration of gonadotropins. 
  4. In Section 6 (ICSI)- Page 7, line 265- Please explain what "Old mice" mean in this context.
  5. Also in Section 6 (ICSI), Page 7- Would suggest a Summary statement at the end of the section to summarize the arguments for and against an effect of ICSI on epigenetics and provide the authors' interpretation of the literature

Author Response

Thank you very much for reviewing our paper and for your comments, which help us to improve the manuscript.

Please found here attached our answers.

Best Regards

ROMUALDO SCIORIO 

Reviewer 3 Report

I had a very positive impression about  Sciorio and El Hajj review. Interesting and well-documented information is provided regarding the epigenetic risks of MAR procedures. The literature considered is worth considering and the final manuscript represents a good summary of the current evidence.
I have only a few minor comments to improve the quality of the review.
- I believe that the acronyms MAR and ART in this context can be assimilated and I therefore suggest using only one
- there is a risk that the tone of the review appears too negative compared to the real impact of the risk of imprinting on the health of the newborn. Although the relative risk is notoriously high, I suggest highlighting, possibly in a small new paragraph, also the absolute risks or the incidence of pathological events.

- Figure 1 does not seem very explanatory. It should probably be reviewed, possibly highlighting which critical steps are involved in the different MAR procedures (in vivo insemination, IVF, ICSI, PGT ...)
- In the paragraph "conclusions" no further epidemiological papers (lines - 462-470) should be cited. Rather, these works could be included in the paragraph that I suggested to include before on the "real" incidence of pathological events.

Author Response

Thanks very much for the time you spent to review our paper. 

Thanks also for the comments, which help us to improve our manuscript.

Here you have attached the replay at these points.

Best Regards

Romualdo Sciorio
